# Lost Heritage—Architectural Replacement of an Atrium and a Courtyard of the Roman Houses of Armea (Allariz, Ourense)

**Marta Lago Cerviño, Adolfo Fernández Fernández \*** , **Alba Antía Rodríguez Nóvoa and Patricia Valle Abad**

Department of History, Art and Geography, Faculty of History, University of Vigo, 32004 Ourense, Spain; mlcervinho@gmail.com (M.L.C.); alba.antia.rodriguez.novoa@uvigo.es (A.A.R.N.); pvalle@uvigo.es (P.V.A.)
\* Correspondence: adolfo@uvigo.es

**Abstract:** Francisco Conde-Valvís's so-called "stone treasure" is a set of unique carved stone pieces, such as bases, column shafts, a mortar, and decorated fragments (trisqueles and rosettes), found during the 2018 excavation campaign in the Cibdá de Armea (Allariz, Ourense). They had been piled up and re-buried—no records existed as to where—at the western end of the Finca de A Atalaia, which was excavated in the 1950s under the direction of Conde-Valvís and began to be excavated again in 2011. The thorough review of the graphic and textual material available from the old excavations allowed us to determine the original archaeological context of the pieces. Most of these elements belonged to the atria of the so-called "Domus of Hexasquel" (North house) and "Domus of the Rosette" (South house). Once we established the origin of all the elements, especially with the aid of the old photographs, it was decided to reintegrate them into the site, to increase the educational and interpretive value of Armea, instead of burying them indefinitely in the warehouses of a museum.

**Keywords:** conservation–restoration; musealisation; lithic elements; Roman sites; Galicia

## 1. Introduction

The 2018 excavation season at the western end of the Atalaia sector, in the Roman site of Armea, led to the discovery of a significant assemblage of stone fragments, including columns, millstones, and decorative elements. Following the initial confusion, these pieces were eventually identified as the stone elements found by Francisco Conde-Valvís during his excavations in the 1950s, which had been initially stored near the perimeter wall of the estate and which, over time, had become interred and forgotten, as no extant records existed about their whereabouts. In 2019, the state of preservation and interpretation of these pieces was thoroughly reviewed, and the texts, plans, drawings, and photographic records of the old excavations were exhaustively examined, in order to establish the original archaeological context of these elements, which were collectively referred to as "Conde-Valvís's stone hoard". After determining their archaeological context, the restoration team was able to put them back in their original position, making the site easier to understand for visitors. The main objective is to present the process that led to the restitution of these pieces to their original location at the site. The result of our intervention fills a gap in the archaeological research and publication about this type of archaeological and conservation–restoration processes in northwestern Spanish sites.

A complete analysis of some parts of Roman domestic architecture (atriums and courtyards) in a small city is presented. This subject has been studied in north Portugal and Asturias, named in Bracara Augusta [1,2], or in other small cities such as Tongobriga [3], Alvarelhos [4] and Chao San Martín [5] with an indigenous origin and with a domestic architecture similar to Armea. However, there was a manifest lack of this type of work in the actual Galicia (NW of Spain) that this paper aims to fill.

To reach these objectives, a multidisciplinary methodology is applied, combining historical archivist, archaeological, and conservation–restoration research based on international principles.

## 2. The Roman Site of Armea

Cibdá de Armea, also known as Hillfort of Armea, is part of the historical–archaeological complex of Armea, which spreads around the eponymous hill, 15 km distance from Ourense, northwest of Spain (Figure 1). In the Roman period, this area was in the northwest of the *Conventus Bracarensis* (ancient Roman province of *Gallaecia*). The hill is part of a low mountain range (average altitude of 600 m. a. s. l) that separates the Miño River basin to the north—specifically the A Rabeda valley, tributary of the Miño—and the Arnoia valley to the south. The complex (Figure 2) comprises a wide variety of archaeological sites, spanning the pre-Roman period and the Late Middle Ages. The most significant sites are the Roman city of A Cibdá, the Basilica of the Ascension and Forno da Santa [6–10] (pp. 77–100), the Roman road-*Camino de Santiago*, and the Late Medieval settlement of Santa Mariña [11], as well as a large number of scattered stone finds (e.g., warrior torsos, triskelia, severed stone heads) [8] (p. 24). The Roman settlement of Señoriño, situated at the foot of the historical path leading up to the Cibdá, has recently been excavated and musealised [9,12,13]. This path currently links the *Camino de Santiago* and the *Vía de la Plata* and is probably a fossilised secondary Roman road running from Chaves (*Aquae Flaviae*) to Lugo (*Lucus Augusti*).

Cibdá de Armea is situated on the hill's northern slope. In the 1950s, a series of Roman courtyard houses were excavated in a terrace known as A Atalaia, flanking a stone-paved street [14]. Systematic excavation of this area was resumed in 2011. Initially, the excavation focused on re-excavating the 1950s trenches, but from 2014, the stratigraphic excavation of new areas began under the direction of a team from the University of Vigo. To date, this has resulted in the full excavation of two atrium houses and the partial excavation of another one. This house type is rare in Roman sites in rural contexts in northwest Spain (Figure 3). One of the earliest occupation phases, from the late 1st to the late 2nd or early 3rd centuries AD, was eminently Roman in character. Thus far, the stratigraphy suggests that there is a hiatus in occupation between the Late Roman and the Late Medieval period. An earlier occupation phase (early to late 1st century AD) has been attested in A Atalaia, which is coetaneous to the construction of Monte do Señoriño [15], as revealed by the ceramic record—Italic and south-Gallic *terra sigillata*, Baetican amphorae, indigenous wares—and razed architectural features under the large houses [12,16]. The analysis of the ceramic contexts found during the recent excavations has allowed us to outline the occupation sequence of Armea [12,16]. Everything suggests that this terrace was first occupied in the early 1st century AD, contemporaneously to the construction of the area of O Señoriño on the site's outskirts [10]. Evidence for this period includes a series of domestic and industrial (metal smelting) features found under the house of the Hexasquel and the main thoroughfare, which have been dated, based on the ceramic assemblage, to the final years of Augustus's or the beginning of Tiberius's reigns [16] (pp. 864–866). The original Roman constructions were modified in the mid-1st century AD and eventually razed to the ground with the re-urbanisation of the sector surrounding the streets, where new large houses were built, probably in the late 1st century AD, during the Flavian period. The area was abandoned between the late 2nd and early 3rd centuries AD, for reasons that remain unclear [16] (pp. 867–871).

The complex of Armea is surrounded by legends and vernacular traditions. Ritual activity is attested at the site before the Roman period, and over time, legend became one with the archaeological remains. Traditional accounts relate the remains to the martyrdom of Saint Mariña, a 2nd-century legend. According to these accounts, Mariña was burned in the sauna beneath the basilica; the "Pioucas" are a former Roman wine press where Mariña went to take a fresh bath after her martyrdom, etc. [17].

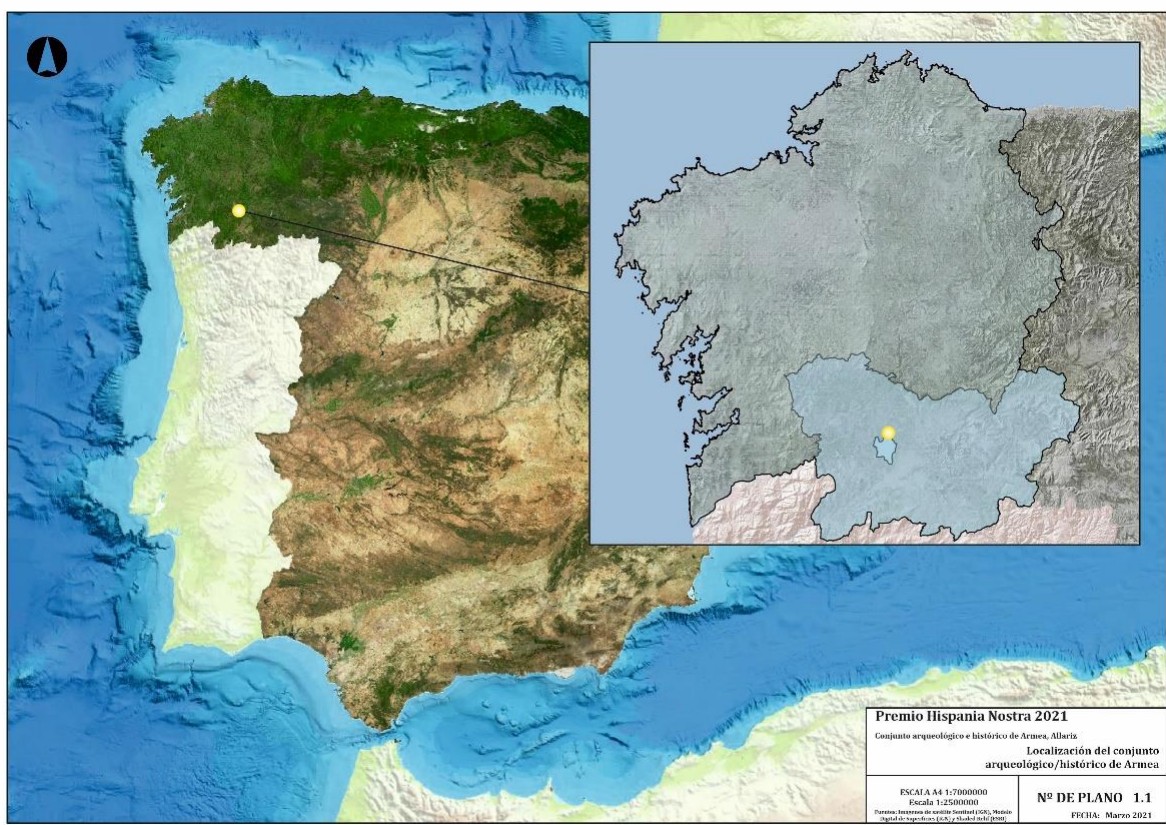

**Figure 1.** Location of the historical–archaeological complex of Armea. Sources: IGN and ESRI.

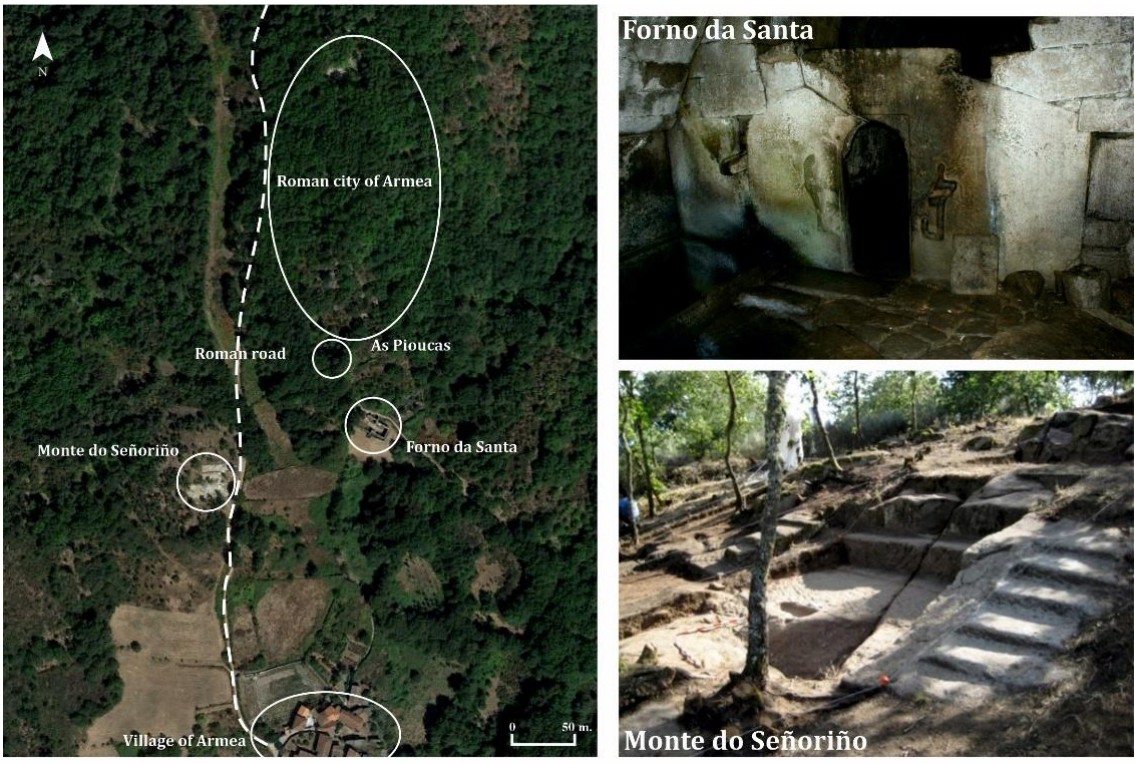

**Figure 2.** Elements in the historical–archaeological complex of Armea.

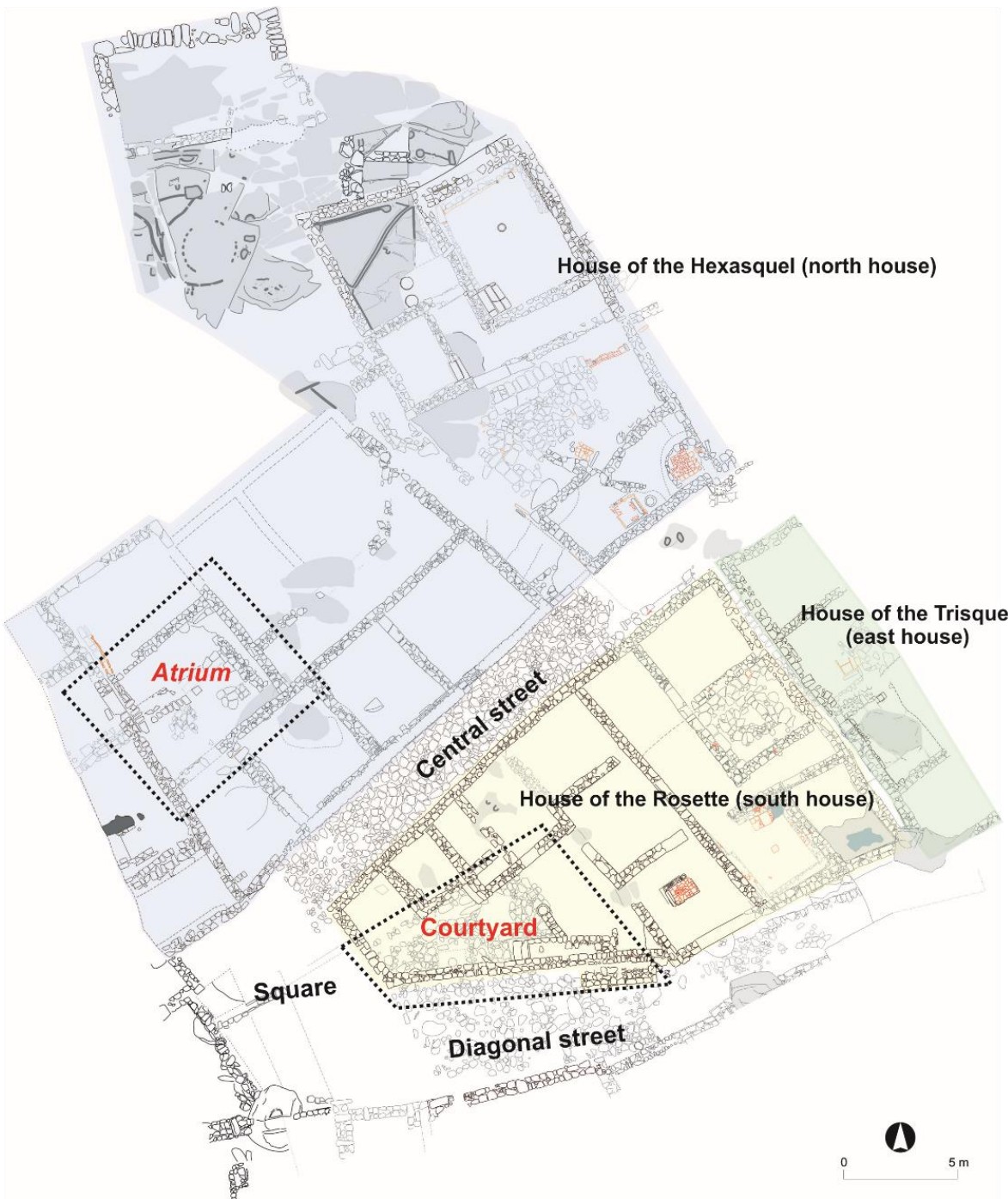

**Figure 3.** Plan of the excavated Roman settlement in A Atalaia.

## 3. Conde-Valvís's Excavations

In the early 1950s, the Allariz-born engineer and amateur archaeologist Francisco Conde-Valvís Fernández undertook, following his father's footsteps, the study of Santa Mariña and Armea. He surveyed, recorded, and collected numerous archaeological remains and even opened a few soundings [18] (pp. 44–46). He has discovered the most substantial and best-known remains found to date in Armea, such as the warrior torsos, the "severed heads", and a large number of decorative stone pieces, such as triskelia, hexasquels, and rosettes, which are currently held in the Archaeological Museum of Ourense (Figure 4).

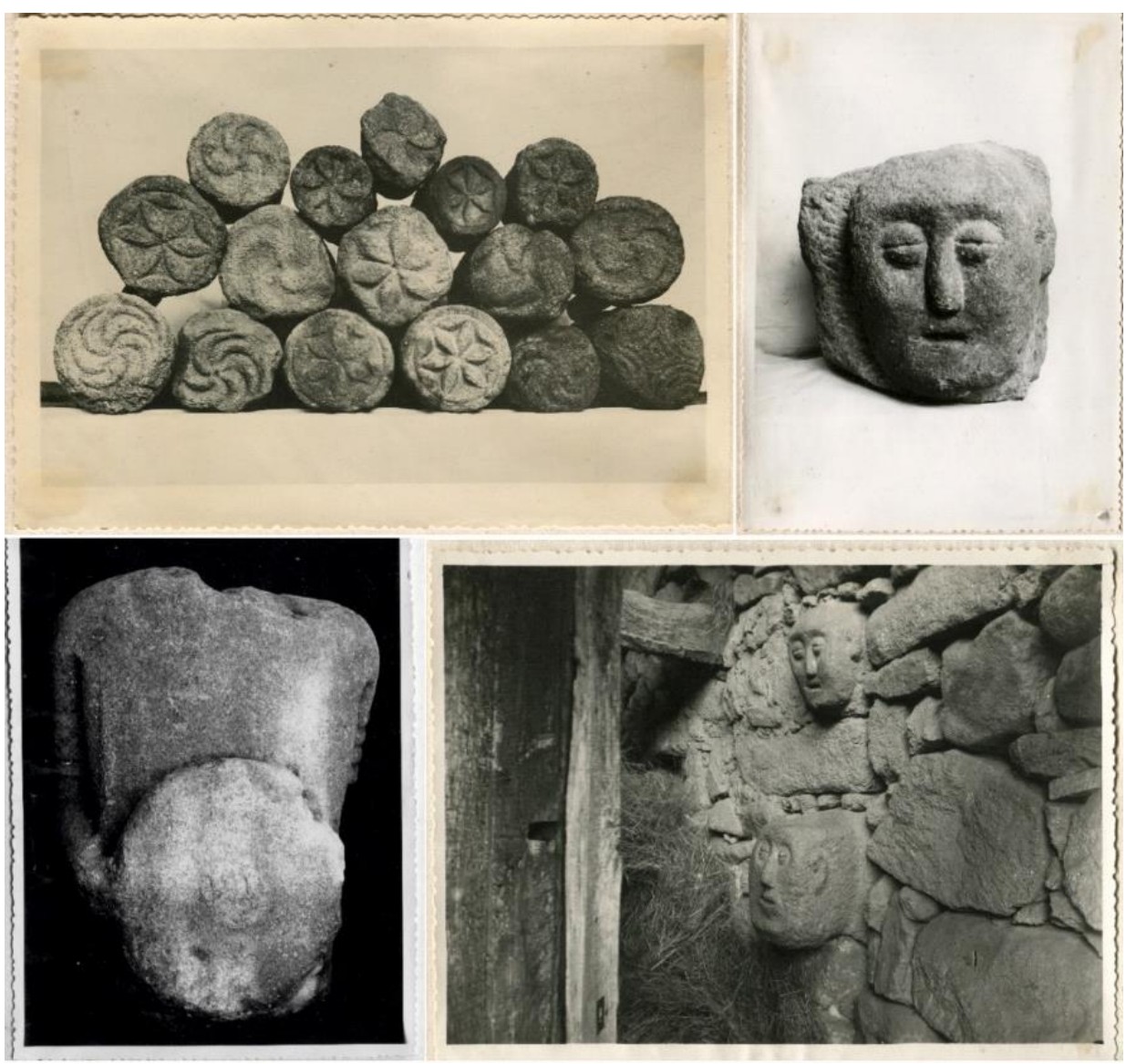

**Figure 4.** Photographs of some of the remains found in the 1950s [7] (Plates V, VIII, XXIII, XXVIII, and XXIX).

He began his first open-area excavation in 1955 in what is today known as "Atalaia". He interpreted the finds as the remains of two Roman atrium *villae* separated by a central stone-paved street. His work led to two publications [14,19], which provide details of the materials found and the structural remains, including numerous plans and photographs. In the course of this work, Conde-Valvís also created a large photographic archive, which is a key source for the archaeological study of Armea. Currently, the archive is held by his family and the Archaeological Museum of Ourense.

After the 1955 excavation, the remains were backfilled and covered by the autochthonous forest, and there they remained for 56 years until the archaeologists returned in 2011.

## 4. Discovery of the Stone Elements

In 2018, during the excavation of so-called Sector 14, at the western end of the northern domus (of the Hexasquel), near one of the atria, a fill associated with numerous carved stone pieces was identified (Figure 5); it was clear that the stone fragments had been piled up and interred deliberately during the 1950s. This find caught the excavators by surprise, as no records existed of this deposit. As they were not included in the collection held

at the Archaeological Museum of Ourense, like the other remains, it was thought that they had been stolen, leading to a complaint of the director of the excavation, Francisco Conde-Valvís.

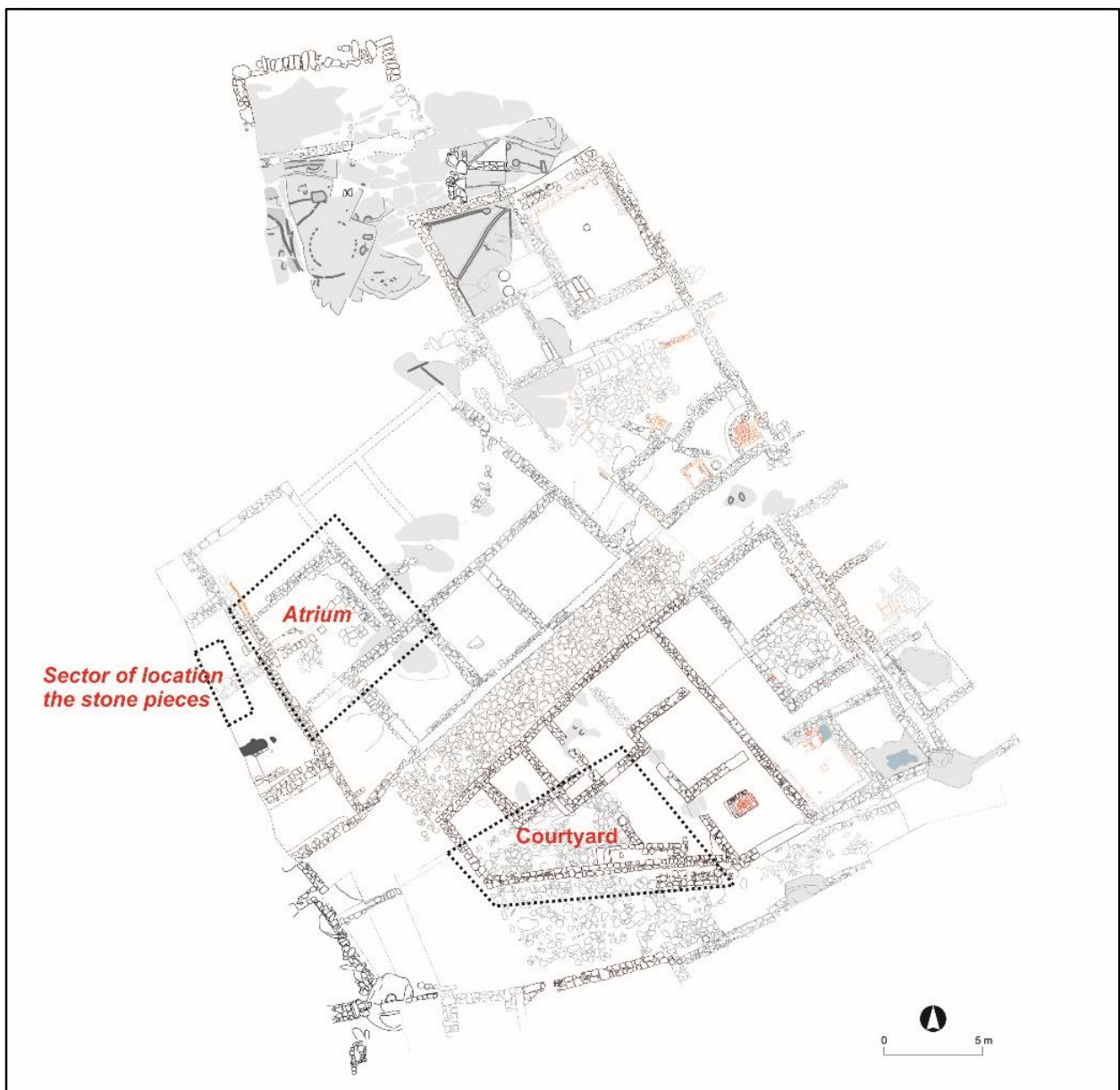

**Figure 5.** Plan of the site with the location of the finds.

When they were found and, especially, after their origin was established, the University of Vigo team began referring to this assemblage as "Conde-Valvís's stone hoard", to highlight the exceptional value of the pieces. The assemblage includes four bases, two column shaft fragments, a capital, the fragment of a possible millstone, a stone mortar, and several fragments of decorative rosettes and triskelia. Until 2019, the assemblage was securely held in a warehouse owned by the City Council of Allariz (Figures 6 and 7). Once the original location of the pieces was determined, they were returned to the site.

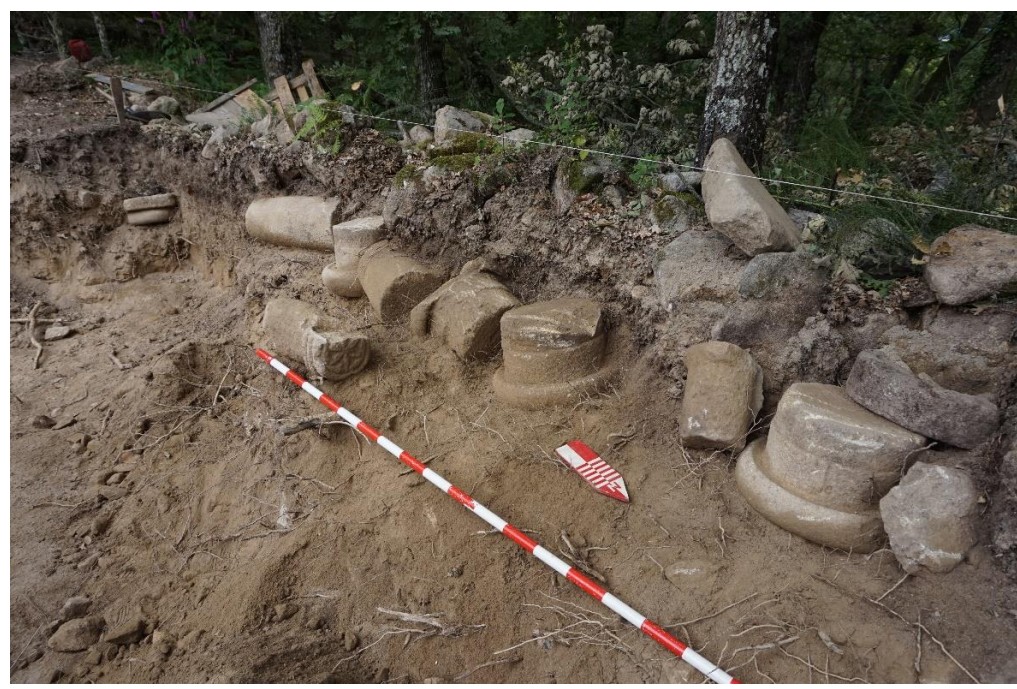

**Figure 6.** Discovery of the stone fragments.

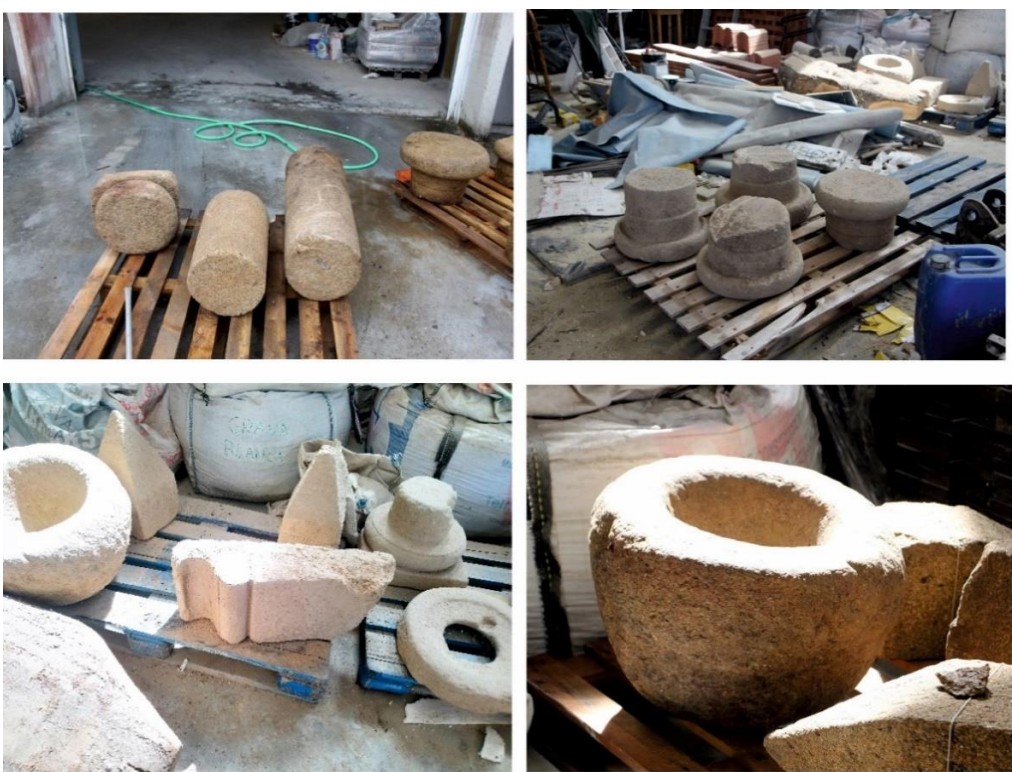

**Figure 7.** "Conde-Valvís's stone hoard"stored in a City Council of Allariz warehouse.

## 5. Identification and Archival Documentation

The project "Accessibility of the archaeological sites of Armea (Allariz) and Outeiro do Castro (Baños de Molgas)", undertaken in 2019, comprised the musealisation of the site and the installation of information boards and elements to channel and organise visitor flows, as well as the comprehensive analysis of Armea's architectural features. The areas excavated from 2011 were examined as a whole, and the consolidation works undergone by

the features were reassessed. This involved a thorough examination of the literature, one of the results of which was the identification of the original locations of the stone elements rediscovered in 2018. The reports for the 2011, 2012, and 2014 excavation seasons, as well as a number of existing articles on Armea, were especially useful in this regard [14,20].

The elements that could be relocated came from two different sectors: first, the paved atrium in the western area of the northern house (also known as the House of the Hexasquel); and second, the western paved courtyard in the southern house (also known as House of the Rosette). Both buildings were excavated and backfilled in 1955 [14] and re-excavated and consolidated between 2011 and 2014. The stone elements found in 2018 were compared with those depicted in the old photographs and plans.

*5.1. The Atrium of the House of the Hexasquel*

Conde–Valvís [14] (pp. 479–483) describes an atrium with four column bases at its centre, forming a perfect square with 2.5 m sides (Figures 8 and 9). He also found two column shaft fragments among the debris, which Valvís placed over the bases, somewhat randomly, to take a picture of the courtyard. It has been impossible to establish whether a capital was also found, as the text is unclear in this regard. Although at first he writes that none of the four capitals were found, in the same article, he claims that three of the four capitals were missing, so this suggests that at least one of them was found. The atrium was paved with small, unevenly sized, stone slabs, and a channel ran across the room from E to W; a square drain, similar to that which is still visible in the southern domus, was also identified. In his article, Conde-Valvís uses the terms *impluvium* and *compluvium* incorrectly. He refers to the square drains in the atrium and the courtyards of both houses as *compluvium* and to the paved areas as *impluvium*. Interpreting this small drain, used to evacuate rainwater, as a *compluvium* is a mistake, for *compluvia* are the openings left on the roofs of atria to let in light and air, as well as to let rainwater fill the *impluvium*, a pool from which water ran into the house's waterpipe system. In Armea, rainwater entered through this opening, which can be regarded as a *compluvium*, but not into an *impluvium*, but directly onto the paved floor. Rainwater filtered through the stone slabs and was channelled to the rest of the house through the quadrangular drain and a stone channel built beneath the pavement.

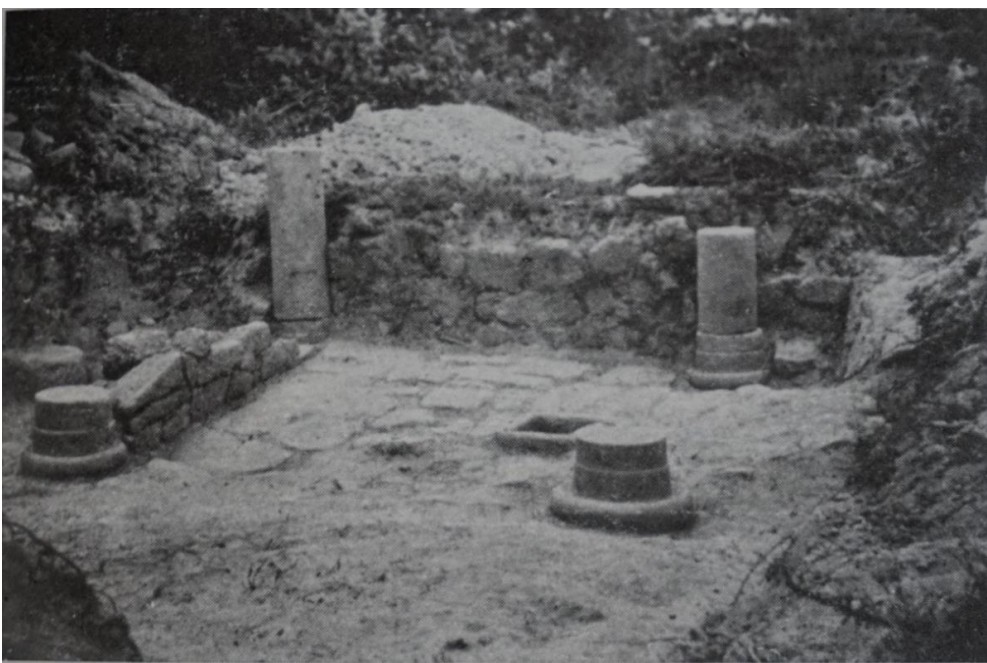

**Figure 8.** Photograph of the atrium after the excavation (Conde-Valvís 1959: Plate I).

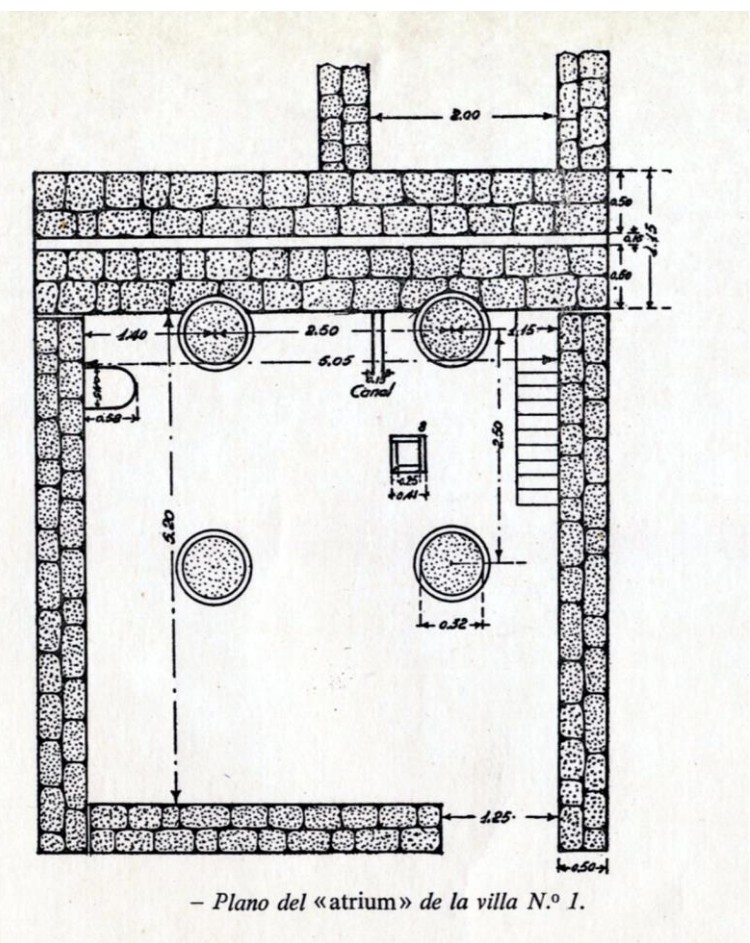

*– Plano del «atrium» de la villa N.º 1.*

**Figure 9.** Plan drawn by Conde-Valvís in 1955 (Conde-Valvís 1959: 482).

Conde-Valvís finishes his description by saying that, a few days after closing the excavation and taking the picture "[ . . . ] a group of vandals walked by, grounded the columns, took the bases out of place, yanked at the slabs [ . . . ], in short, these soulless miscreants left behind them the mark of barbarism" [14] (pp. 481–483). After this incident, the soil beneath the slabs could be excavated. Nothing is said about whether the vandalised elements were destroyed, stolen, or partially recovered. The 2018 excavation revealed that at least some of them had remained on site and had been interred by the excavators at the edge of the property, next to the atrium. At the end of the excavation, the trenches, including this atrium, were backfilled [20].

The consolidation and musealisation works in 2018 included raising the walls by adding a few protective courses and the restitution of part of the pavement and the drain lids, using slabs found among the fills that covered the rooms (Figures 10 and 11). Additionally, in the words of the director of the excavations, "In order to present a discourse that is closer to reality, two fragments of Roman columns found in the vicinity were placed on the bases, conveying an image that helps to reflect the archaeological discourse" [20] (pp. 89–90). After consulting some of the archaeologists, it was concluded that the original location of these column shafts cannot be established with any certainty, and that it is not even clear that they belong to Armea. They were brought to the site in 2014 from the house where they were found out of context in Allariz's historical quarters. The analysis of their raw material (granite), morphology and typology, and their comparison with other columns found in the archaeological context in Armea, strongly suggest that they are foreign to the Cibdá, and they cannot even be securely dated to the Roman period to which the site is

dated. Following this, it was decided to remove these columns, especially after the original column shafts were found in "Conde-Valvís's stone hoard".

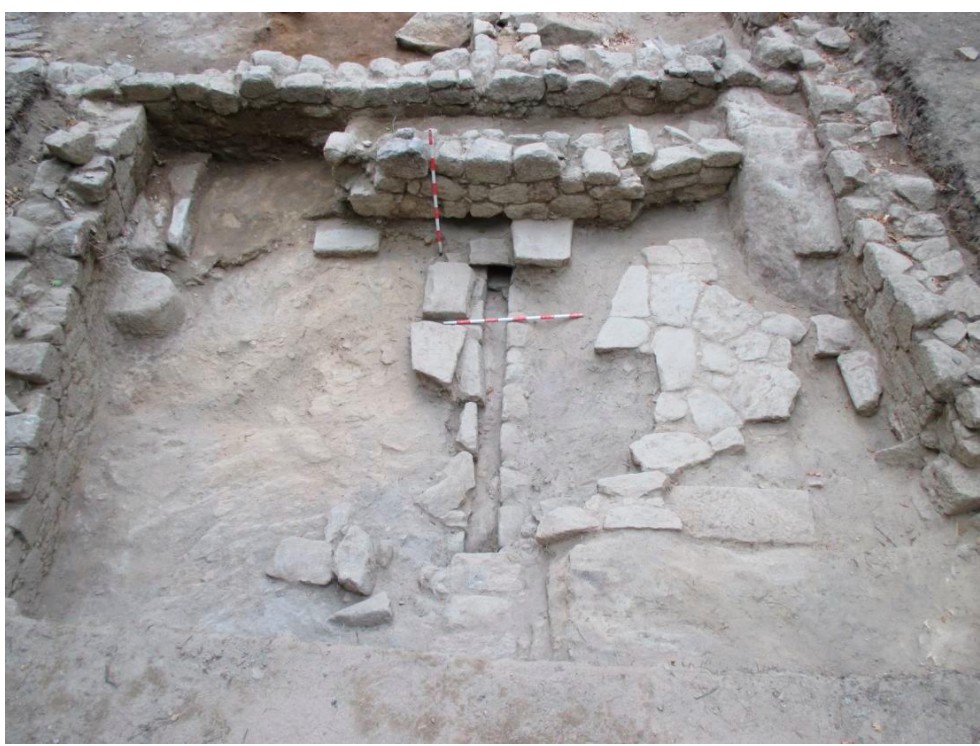

**Figure 10.** State of the courtyard after the 2014 excavation. Photograph: David Pérez.

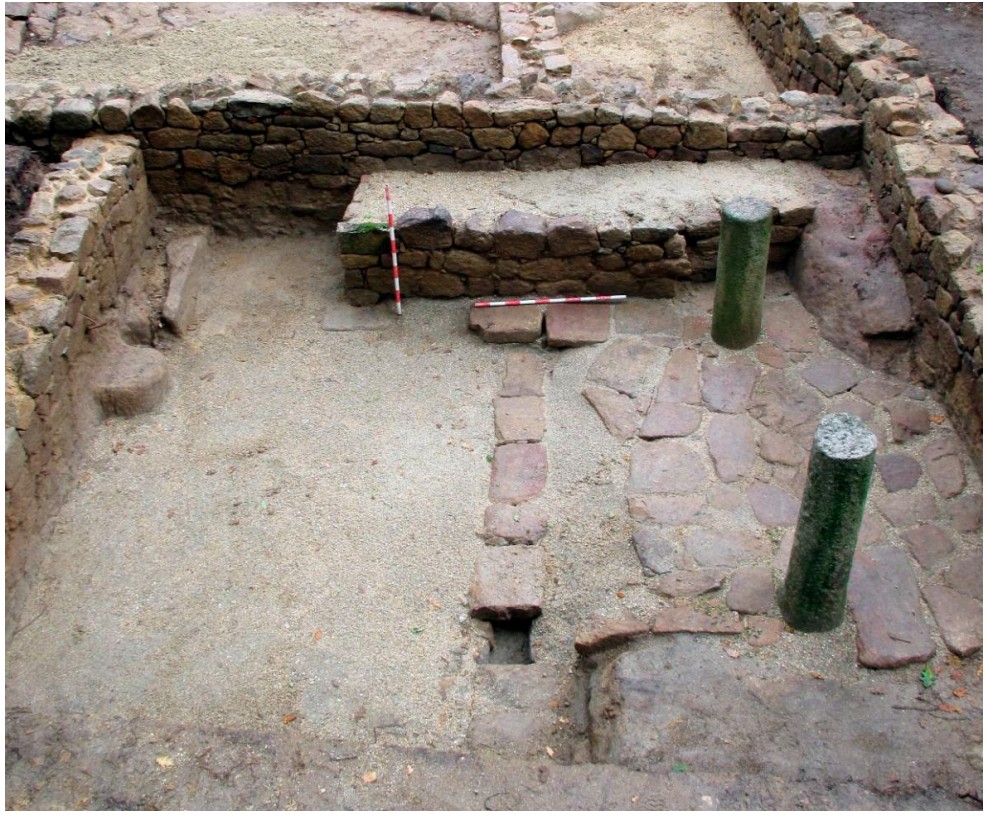

**Figure 11.** Final state of the atrium after the restoration works in 2014. Photograph: David Pérez.

### 5.2. The Courtyard of the Domus of the Rosette

Conde-Valvís [14] (pp. 483–484) describes this courtyard as a floor paved by small, unevenly sized, stone slabs, with a channel running across it, from E to W, and a *compluvium* built with four stones, 0.12 m thick, forming a square deposit with 0.25 m sides (inner measurements). A staircase on the south side of the wall gave access to an upper storey. A small wall projects from the staircase towards the centre of the courtyard. It is 0.80 m long and 0.40 m thick, and it ends in a crudely carved granite block, which is approximately 0.50 m high (Figure 12). In his publication, Conde-Valvís, along with the general arrangement of the courtyard, describes different architectural elements found during his excavation, including the base of a column that probably sat on the granite block and a column shaft fragment which he placed on the base to take a photograph (Figure 13). He also describes a mortar found to the east of the small wall. Finally, he mentions a fragmentary millstone that sat vertically on the channel, acting as a water sluice. Like with the atrium in the house of the Hexasquel, the building was backfilled after the excavation. Again, no information is provided as to the whereabouts of the stone elements found, whether they were interred in situ, or taken to a different location.

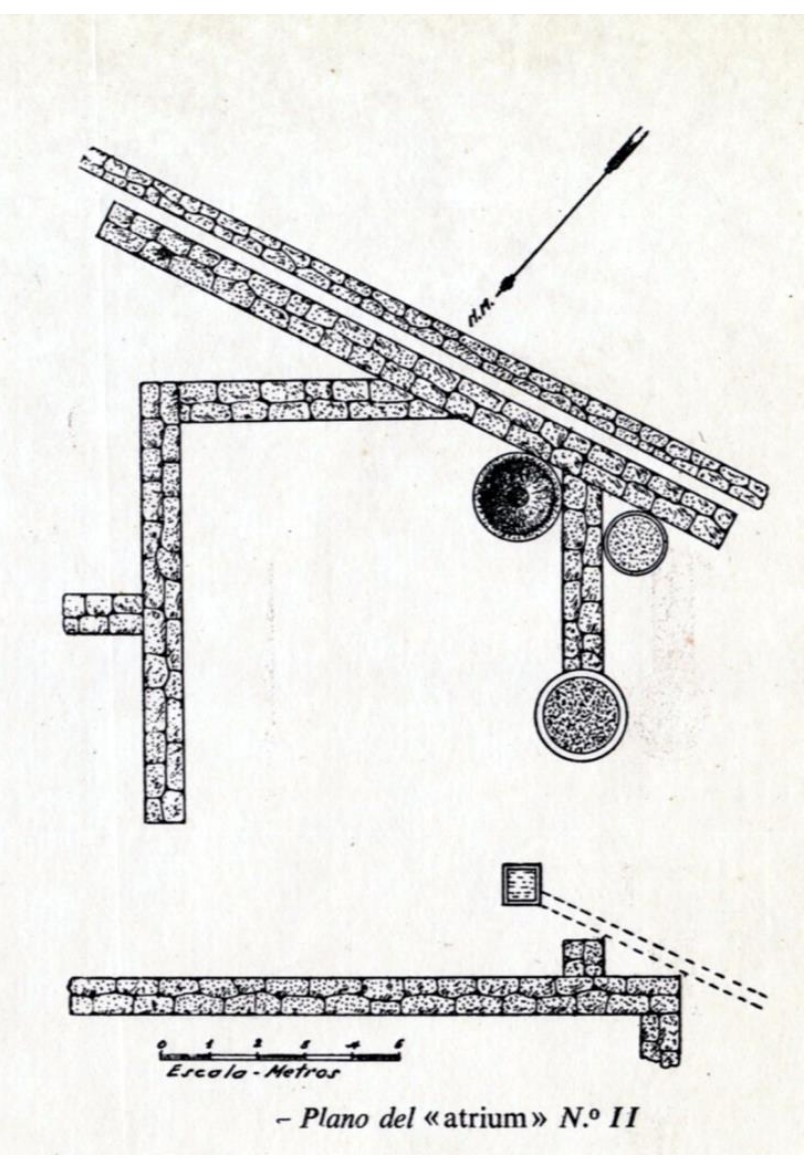

**Figure 12.** Plan drawn in 1955 [14] (Conde Valvís 1959: 486).

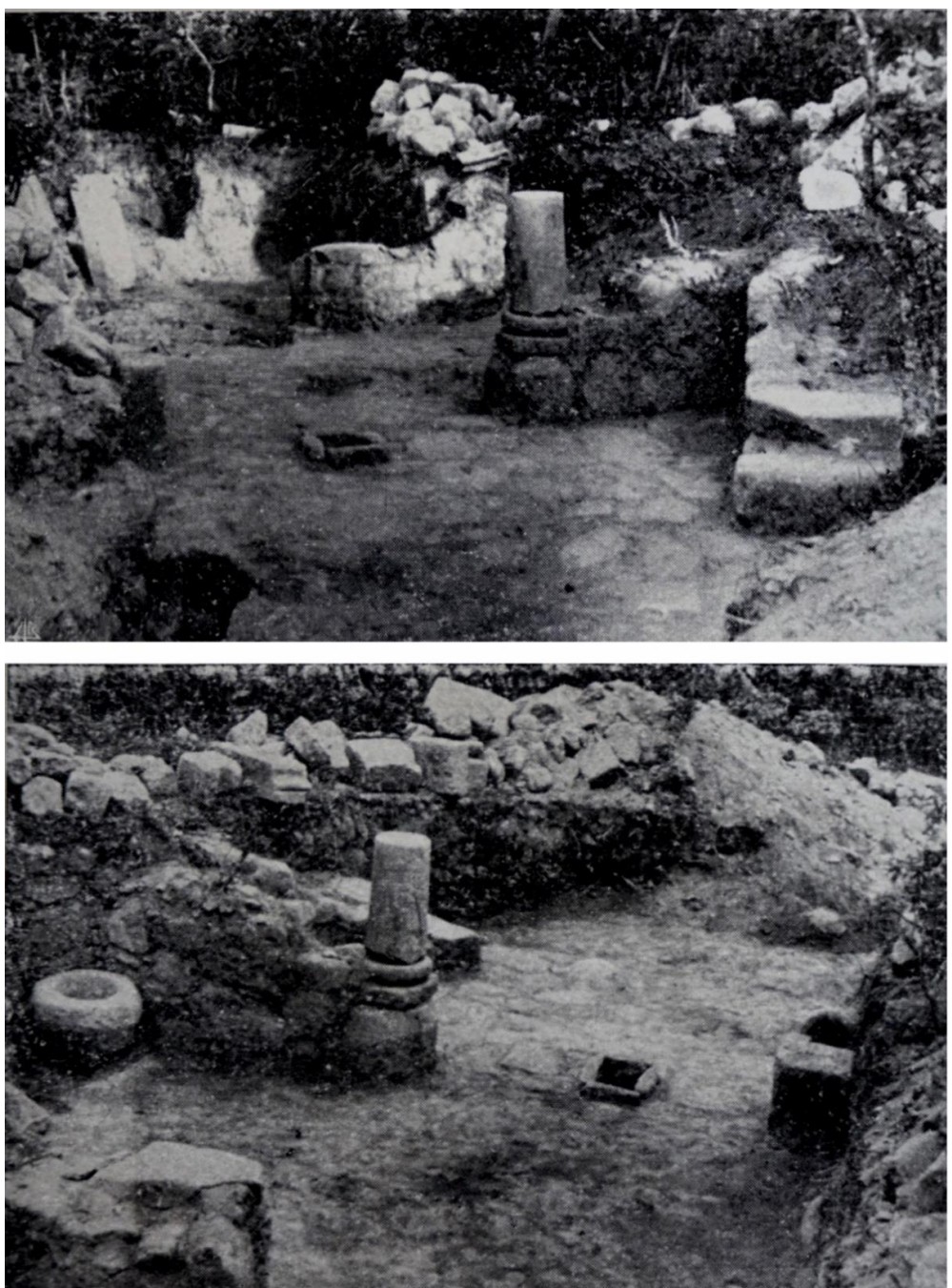

**Figure 13.** Several views of the courtyard of the Rosette after excavation [14] (Conde Valvís 1959: Plate II).

Between 2011 and 2012, the City Council of Allariz resumed the excavation of this area; the excavation was directed by David Pérez. The loose objects described by Valvís could not be found, except for a few threshold stones that had been piled near the site (Figures 14 and 15). The crude granite base was found in situ, and a small column shaft found near the estate's perimeter wall was placed on top, in an attempt to recreate Valvís's old photographs [20] (p. 89).

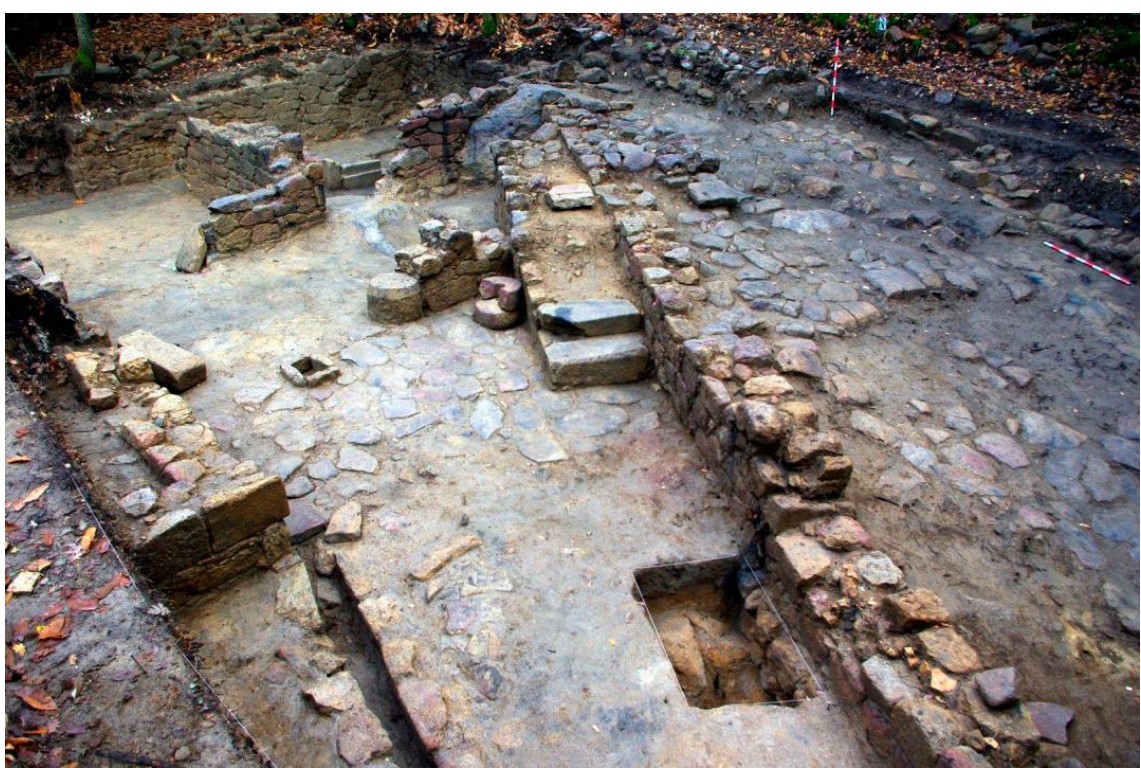

**Figure 14.** View of the atrium of the Domus of the Rosette after the 2011 excavations. Photograph: David Pérez.

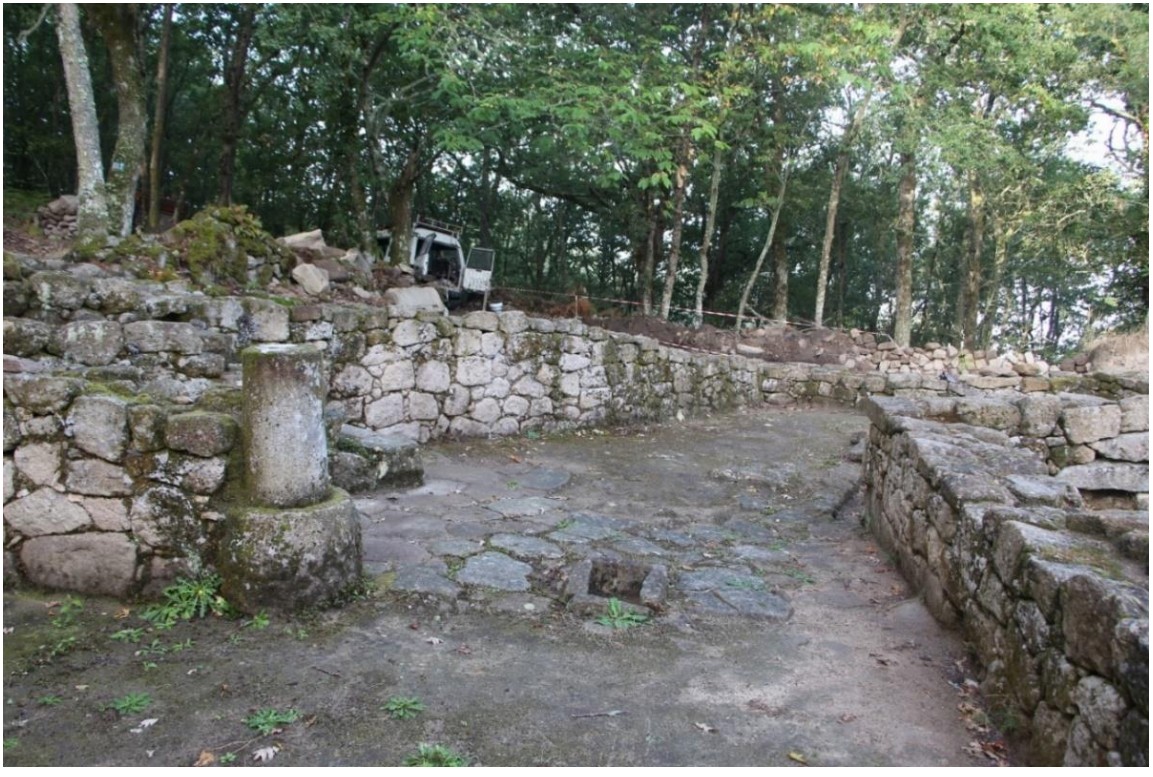

**Figure 15.** View of the site after the 2011 and 2012 works: Photograph: Council of Allariz.

## 6. Results

After the existing records were examined, and the accurate measurement of the pieces in "Valvís's tone hoard" were taken and compared with the 1950s photographs and plans, it was decided to put back all those elements whose provenance was secure in their original location. Works were developed respecting the current preservation–restoration criteria, following the regulations gathered in the International Charters of Restoration, the deontological codes settled by the ECCO, and of course the Heritage laws applicable in the Spanish state and the autonomous region of Galicia [1]. We have followed as basic criteria the regard to the original pieces, the compatibility and safety of the materials, the differentiation of the added parts, and, as far as possible, the reversibility of the treatments.

### 6.1. Work in the Atrium of the Domus of the Hexasquel

Two column shafts erected in 2014 were removed and replaced with the column bases and shafts found in 2018, whose original location was established through the examination of the excavation records. The pieces were carefully removed, and no elements in the vicinity were damaged as a result. In addition, in order to situate the four original column bases in their original position, the small earth banks erected after the 2014 excavation were dismantled from both the pavement and the back wall of the atrium (Figure 16). Before removing these added elements, the original pieces were stained with clay, following the photographs of the state of the atrium immediately before the 2014 excavation.

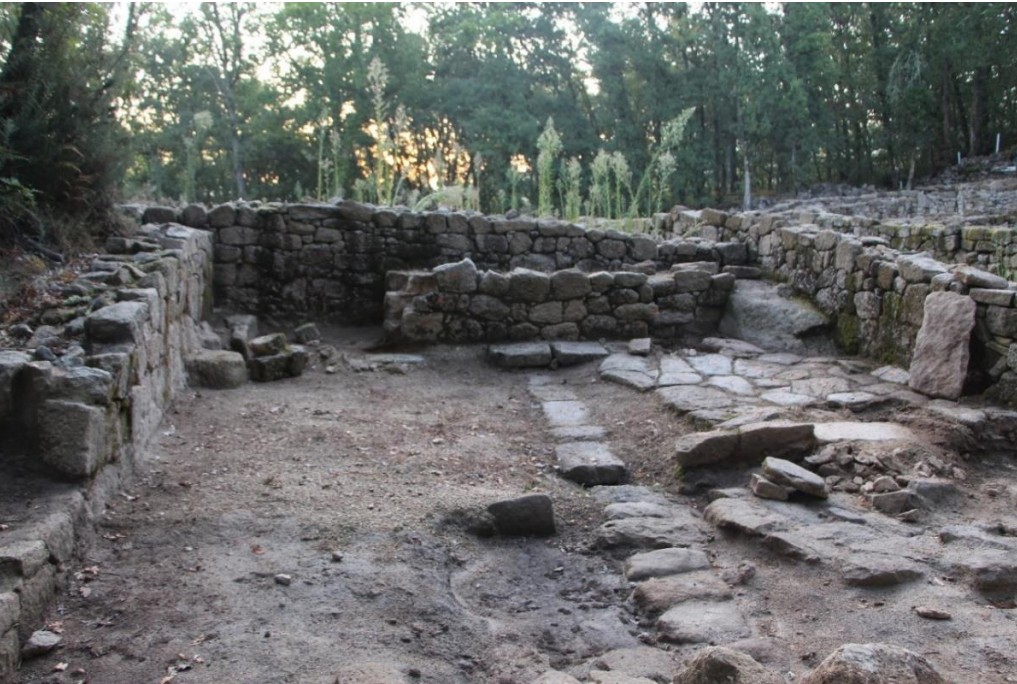

**Figure 16.** The atrium during works undertaken after the removal of the column shafts erected in 2014.

After removing the foreign stone pieces, the four original stone bases were put back in their original location. The analysis of the two column shaft fragments revealed that they belonged to the same column, one of them sitting atop the other, and their lower diameter only fitted one of the bases. One of the old photographs (Figure 17) shows a small wall between the two bases on the north side of the courtyard, but this wall features in no other record, either textual or graphic, so its reconstruction was decided against. Similarly, the drain could not be restored, but its location was marked on the pavement.

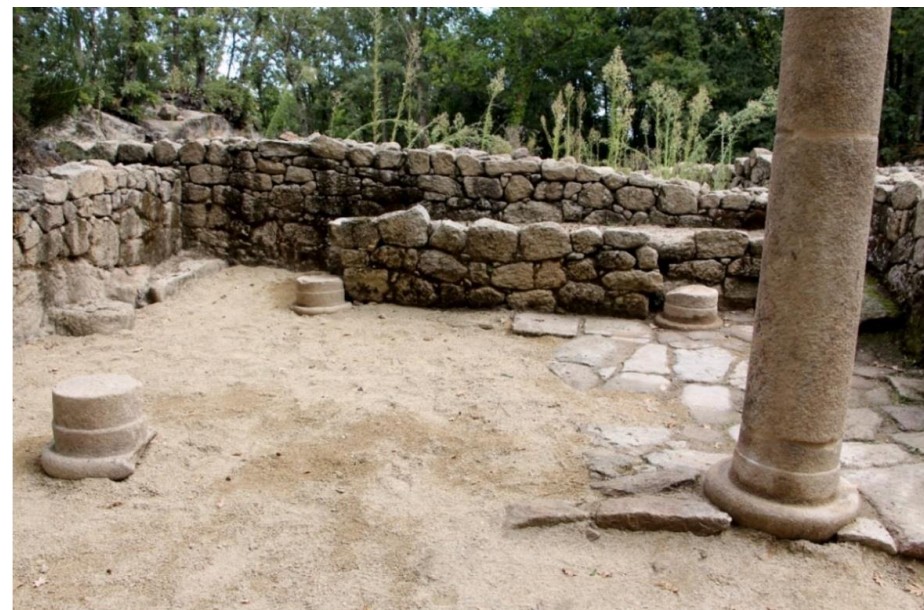

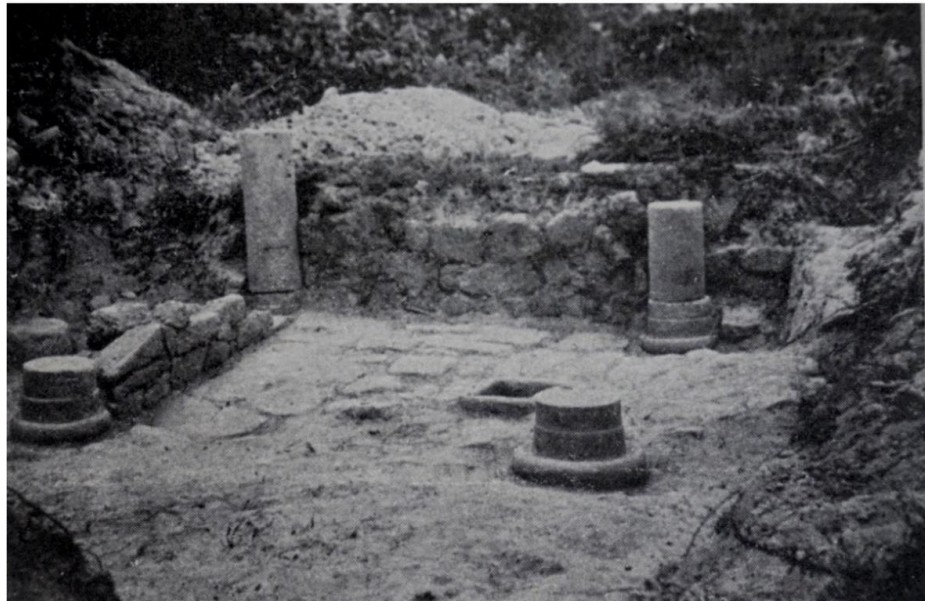

**Figure 17.** Comparison of the atrium in 1955 and after the 2019 works once the stone elements had been restored to their original location and the floor had been levelled.

### 6.2. Work in the Courtyard of the Domus of the Rosette

Two major actions were undertaken in this domus: the opening of an entranceway in the western wall, restoring one of the thresholds identified in the 1950s; and restoring two pieces in the "stone hoard" to their original location, the mortar, and a column base. The millstone was not put back in its original location, as it was thought that it could be easily stolen.

#### 6.2.1. Restoration of the Original Threshold

The wall that closes the courtyard to the west was excavated and backfilled in the 1950s, but none of the old photographs shows its condition at the time. The wall was found again between 2011 and 2012, and during the restoration process, several protective courses were raised. The preserved height of the wall was low, especially in the south, where it barely raised from the ground. With this information, it was thought that this might

have been the location of a threshold, which had been wrongly elevated. The presence of a drainage channel to evacuate water from the house reinforces this hypothesis, as this channel may have run beneath the threshold (Figure 18).

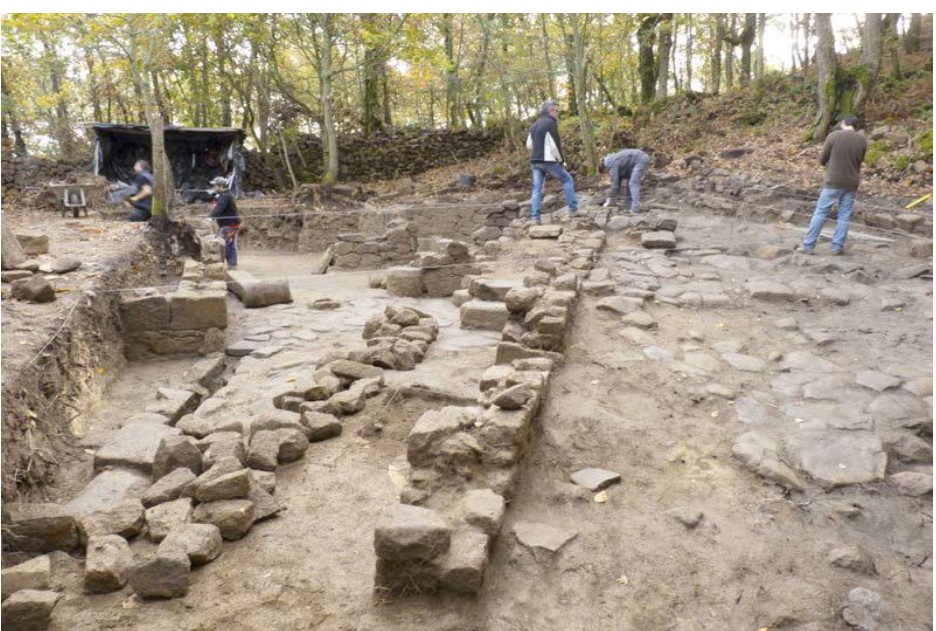

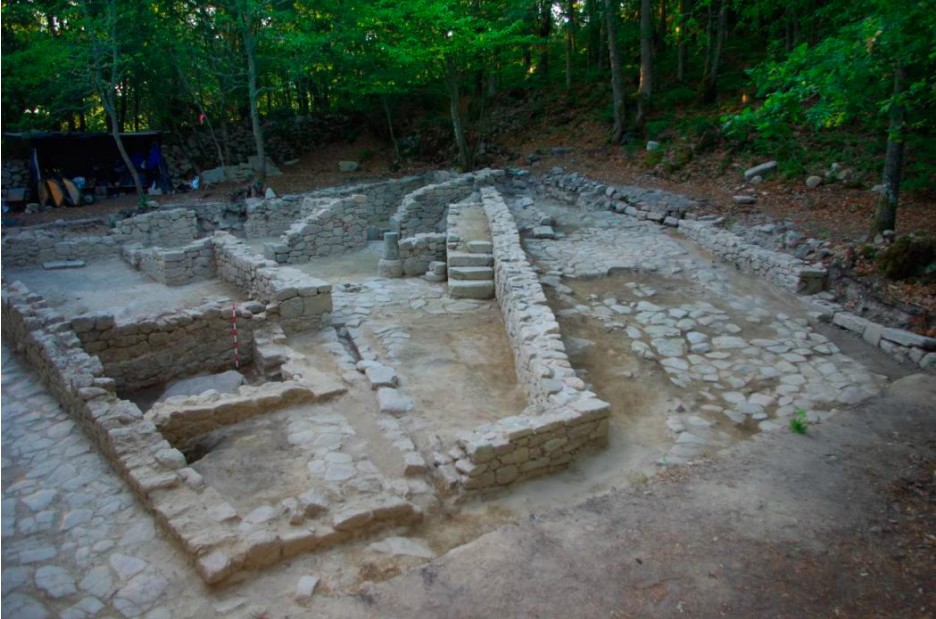

**Figure 18.** Above: view of the western wall in 2011; below: the wall, from the south, after the construction of the protective courses.

In order to open the entranceway, the protective courses were removed from the south end of the wall, always following the photographs and using the schist slabs placed in 2011 to separate the original wall from the reconstruction as guides. After the removal of these additions, one of the thresholds found in the old excavations (Plate VI) was placed on the entranceway [14] (Figure 19). The seat of the threshold was raised slightly to allow the channel to run underneath it. The other doors in the site were used as a model for the reconstruction, which was flush with the house's south wall (Figure 20).

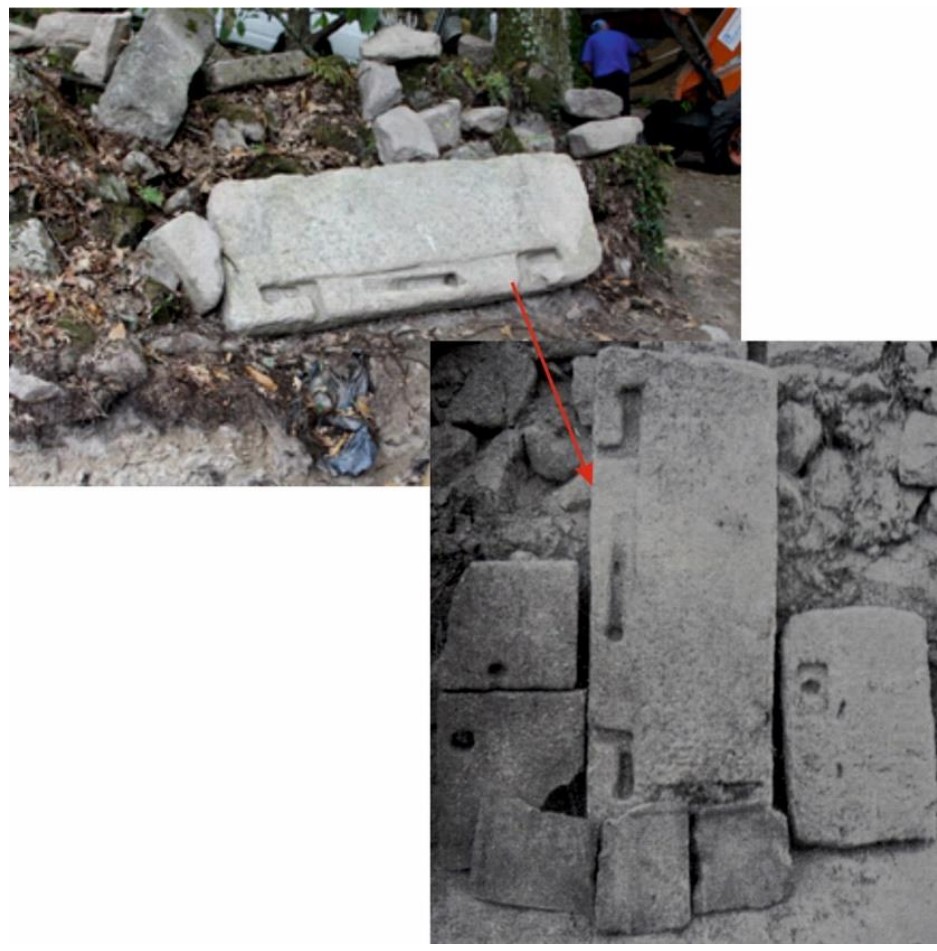

**Figure 19.** Above: view of the threshold in 2019, before it was put in place; below: the same threshold in 1955, when it was found by Conde-Valvís.

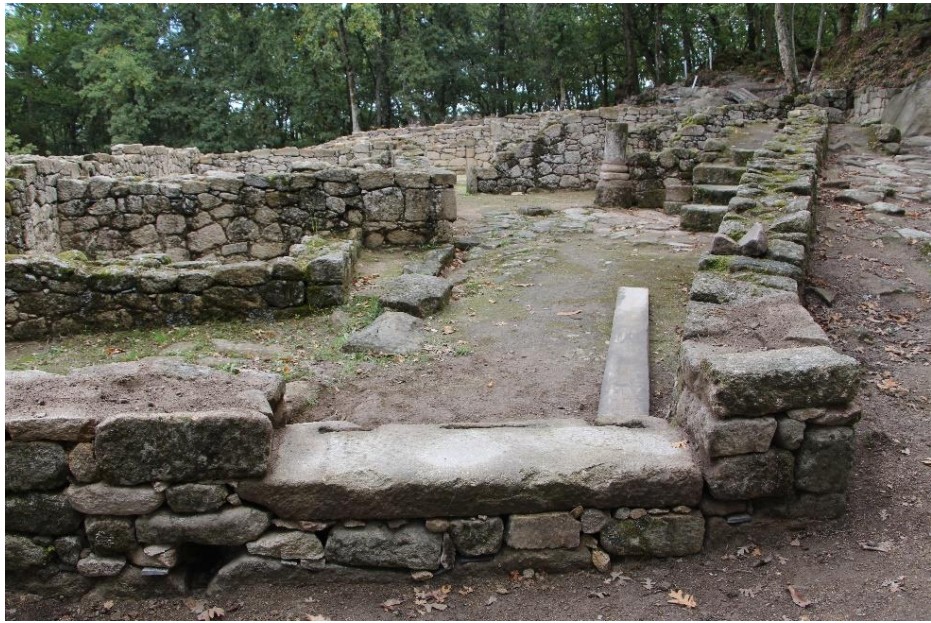

**Figure 20.** Final view of the area after the opening of the entranceway and the installation of the threshold.

### 6.2.2. The Tuscan Capital and the Stone Mortar

The mortar and the capital (maybe reused in Roman times as a column base) were also placed in their original position inside the courtyard [14] (Plates II and IV). It was believed that, owing to its great weight, it was safe to simply place the mortar on its own, without any other form of support, only dug somewhat deeper into the ground. The capital, on the other hand, was bound with lime mortar and used as a base for the column shaft that had already been placed on it in 2011. Although no hard evidence exists to ascertain that it was the one found and placed by Conde-Valvís for the photograph in 1955, it was very similar in terms of size and morphology, so it was thought advisable to place it on the base for the capital's own protection (Figures 21–23).

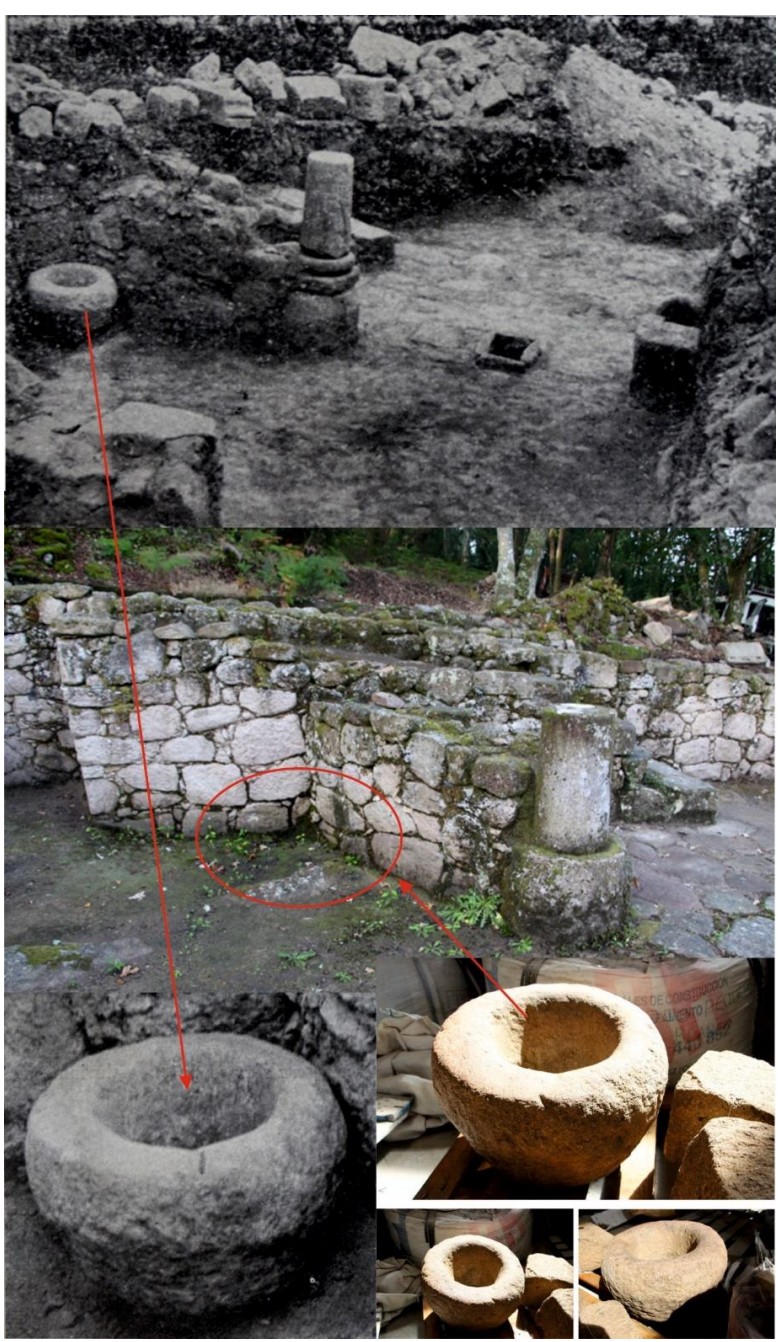

**Figure 21.** The mortar in 1955, when it was first found, and in 2019, before it was restored to its original location.

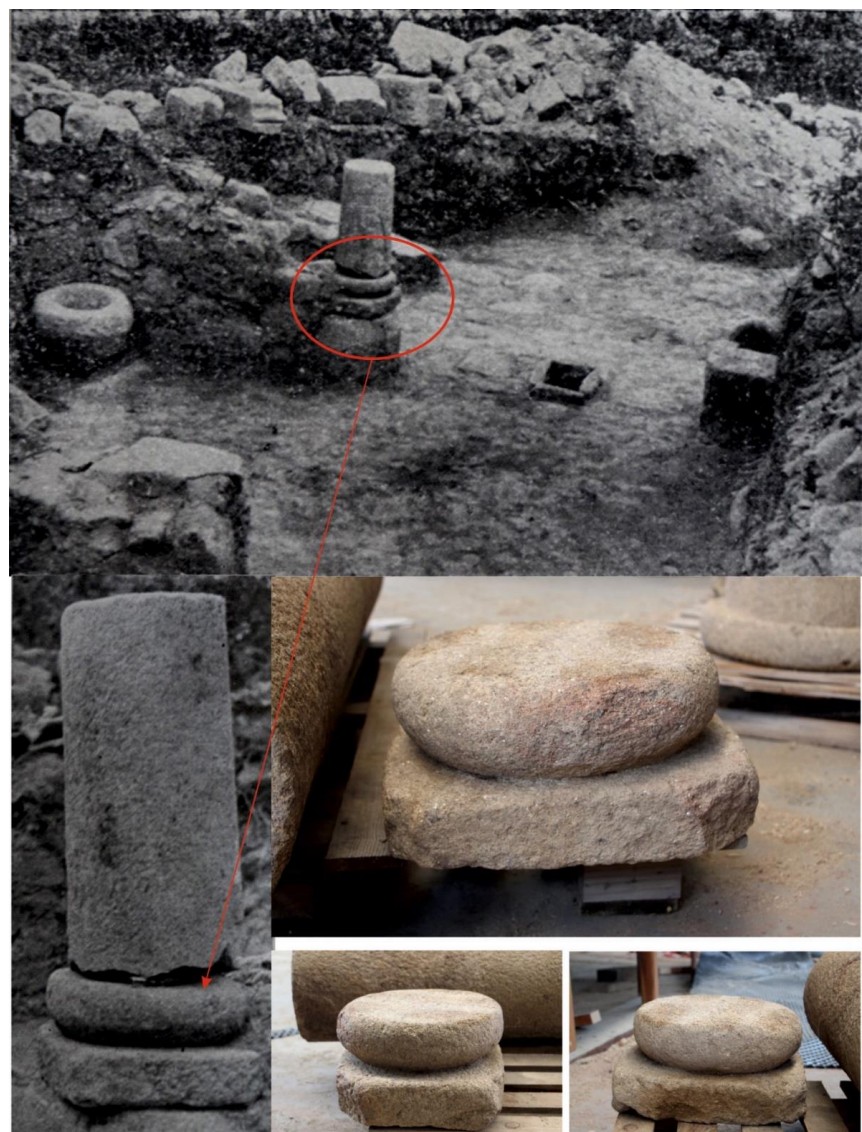

**Figure 22.** The capital in 1955, when it was first found, and in 2019, before it was restored to its original location.

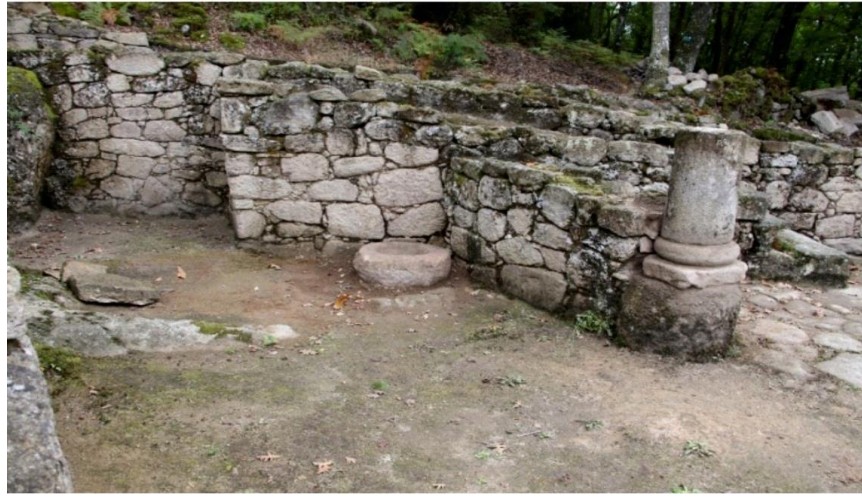

**Figure 23.** Final view of the capital and the mortar in their original location.

## 7. Conclusions

The restoration and musealisation of a site, especially when it has been excavated before, always involves decisions that, inevitably, affect the archaeological interpretation of the remains. In Armea, historically inaccurate reconstructions had been undertaken that undermined the correct understanding of the site. Similarly, although the excavations carried out in the 1950s had left abundant textual and graphic records, the place where many of the elements found had been stored was lost.

In 2019, a comprehensive restoration and musealisation project for the site as a whole, not only to protect it, but to organise visits and ensure that the archaeology was correctly understood by the visitors (over 10,000 per year), was implemented. Two clear avenues of action were established:

- To remove elements, added to the original features in previous preservation/restoration projects, which were not certain to belong to the site.
- To restore stone elements in "Valvís's stone hoard" or other assemblages found at other times and left at the Cibdá for which secure graphic and archaeological evidence existed.

Therefore, the fragments in the "stone hoard" were compared with old photographs to determine which ones could be restored.

Throughout, archaeologists and conservators worked in close cooperation. This was essential also for other tasks undertaken in 2019: the ideal recreation of the houses by an illustrator (Figure 24), within the framework of the musealisation of the site, which necessarily implied rethinking the archaeological remains. The location, identification, and restitution of Valvís's stone hoard allowed for the courtyard and atria to be correctly interpreted and for a previously lacking entranceway into the house of the Rosette to be opened. The restitution of the pieces prevents them from joining a museum collection and disappearing from sight forever, as their limited visual appeal makes them unsuitable for display. In their current location, they are in full view and have didactic value, further increased by the fact that they stand where they were meant to from the start.

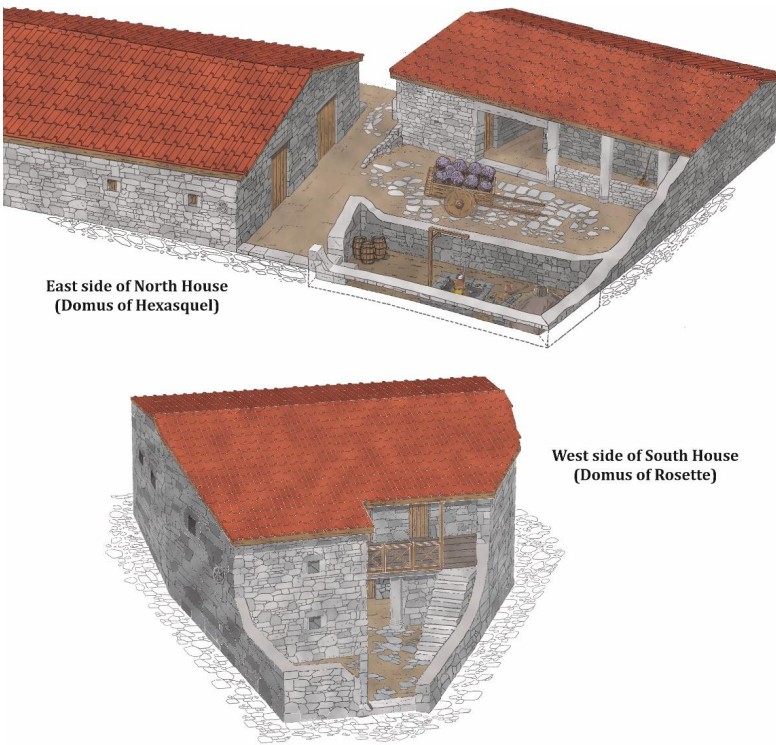

**Figure 24.** Ideal recreation of the houses (Illustrator: Iago Araújo).

**Author Contributions:** Writing and figures: M.L.C., A.F.F., A.A.R.N., P.V.A.; supervision and coordination: A.F.F. All authors have read and agreed to the published version of the manuscript.

**Funding:** This research was funded by the Council of Allariz and Xunta de Galicia. Adolfo Fernández has a Ramon y Cajal research contract. Patricia Valle and Alba Rodríguez both have a Margarita Salas research contract.

**Institutional Review Board Statement:** Not applicable.

**Informed Consent Statement:** Not applicable.

**Data Availability Statement:** Not applicable.

**Acknowledgments:** Iago Araújo has made the recreation of the houses. David Govantes has translated the original text.

**Conflicts of Interest:** The authors declare no conflict of interest. The funders had no role in the design of the study; in the collection, analyses, or interpretation of data; in the writing of the manuscript, or in the decision to publish the results.

## Note

[1] These norms were followed: The Athens Charter for the Restoration of Historic Monuments. 1931; International Charter for the Conservation and Restoration of Monuments and Sites. The Venice Charter. 1964; The Burra Charter. 1979 (revised 1999); Charter for the Conservation and Restoration of Cultural and Arts Objects. Siena. 19871; The Charter of Krakow. 2000; E.C.C.O. European Confederation of Conservator-Restorers' Organisations. Professional Guidelines (II) Code of Ethics. 2003; Lei 16/1985, do 25 de xuño, do patrimonio histórico español; Lei 5/2016, do 4 de maio, do patrimonio cultural de Galicia

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
