# Peer review of "Lost Heritage. Architectural Replacement of an Atrium and a Courtyard of the Roman Houses of Armea (Allariz, Ourense)"

_heritage, doi:10.3390/heritage5010024_

Round 1

Reviewer 1 Report

This is a very interesting work, which deals with a set of carved stone pieces from the Roman city of Armea (Galicia, Spain), found, first, during archaeological campaigns in the 1950s and rediscovered in the 2018 excavation. This work reveals the importance of interdisciplinary approaches in the restoration and musealization of archaeological sites (archaeologists, conservators, geologists, illustrators …) and underlines the value of preserving the artifacts in their original location.

The manuscript is well written, clear, organized and presents a strong illustrative component. Given the above, I am definitely in favour of its publication, with some minor improvements. Please, consider the particular comments listed below:

1 Figure 1/ p. 2: [graphic issues]

The point of location of the archaeological site, in the two maps, and the line connecting them, are not very clear. Maybe it would be possible to highlight them.

2 Figure 2/ p. 3/ (59): [punctuation]

In the legend of the figure, the final full-stop is missing.

3 Figure 2/ p. 3: [graphic issues]

Would it be possible to resize the bottom right picture, according to the size of the picture above?

4 “2. The Roman site of Armea”/ p. 3/ 3rd paragraph/ line 4 (89) [typography]

“a 2nd-century legend”, verify, please, if the dash between “2nd” and “century” should be kept.

5 Figure 3/ p. 4 [graphic issues]

Most of the items shown in the picture begin with a capital letter (Atrium, Square…), except the references to the houses (house of…). Is it purposeful?

6 “3. Conde-Valvís´s excavation”/ p. 4/ 1st paragraph/ line 2 (98/99) [sentence construction]

On the second line of the first paragraph, a change is suggested, in order to avoid the repetition of “found”.

7 Figure 4/ p. 5/ (111): [punctuation]

In the legend of the figure, the full-stop is missing.

8 “4. Discovery of the stone elements”/ p. 6/ 2nd paragraph/ (125-131) [content]

Although it is not the purpose of this work, I would like to ask if it would not be pertinent to say something more about the architectural elements, namely the capital, in terms of typology/ dimensions.

You mentioned that the capital was reused as a base (point 6.2.2./ p. 19). I wonder if it could have been the initial function, as with many Tuscan capitals/bases?

9 Figure 9/ p. 9: [graphic issues]

In the downloaded file, it wasn´t possible to see the image. Verify this point, please.

10 Figure 9/ p. 9/ (185): [punctuation]

In the legend of the figure, the full-stop is missing.

11 “5.1. The atrium of the House of the Hexasquel”/p. 10/ 3rd paragraph/ line 7 (192) [spelling]

“After consulting some of the archaeologist...”. Verify, please, if “archaeologist” should not be in plural form.

12 “5.2.The courtyard of the Domus of the Rosette”/ p. 11/ title/ (205) [typography]

The space between the numbering and the text is missing.

13 Figure 15/ p. 14/ (236) [punctuation]

The colon between “Photograph” and “Council” is missing.

14 “6.2.1.Restoration of the original threshold”/ p. 16/ title/ (271) [typography]

The space between the numbering and the text is missing.

15 “References”/ item 1/ p. 23 (359) [typography]

The first letter of “Evolución” is not in italics.

Author Response

Dear Reviewer 1, 

Thank you for your job and well-pointed comments. We have introduced all modifications and corrections proposed. 

Reviewer 2 Report

The paper is about the re-discovery of the so-called stone treasure by Francisco Conde-Valvìs, a group of carved stones discovered in the 1950s during the excavation of Cibdà de Armea (Allariz, Ourense), of which the scholars had lost track. New excavation campaigns on the same site led to the casual re-discovery of the fragments and the researchers started a detailed work of comparison and analysis of the previous excavation documentation to understand the original position of each stone fragment. Subsequently they managed to re-locate all the elements to improve the scientific meaning and understanding of the Armea archaeological site.

The manuscript is clear, relevant for the musealization of archaeological sites sector and presented in an appropriate manner.

The cited references are appropriate and well-balanced.

The manuscript and the research are scientifically sound. The figures and images appropriately support the explained methodology and they are easily understandable. The conclusions are consistent with the evidence and arguments.

Author Response

Dear Reviewer 2, 

Thank you for your job and revision. 

Reviewer 3 Report

The paper, submitted for peer review, has seven essential parts, including an introduction, five main body paragraphs, a conclusion, and a brief bibliography. The introduction provides information related to the discovery of architectural elements and details their rediscovery in the recent past and their use during conservation work. Unfortunately, the brief introduction does not contain information about topic importance, the study aims and their significance, and knowledge gap of the investigated theme. The authors did not introduce research methods. The second paragraph provides information on the geographical location of the site, archaeological data and some historical facts.

The third section of the paper presents the history of research. The two next parts inform about archival research and lapidary research in 2011, 2014, 2018-2019, resulting in uncovering of the location of architectural details and elements discovered in 1950-1955. And finally, the last section is dedicated to the conservation process on the site and demonstration of the project results. In conclusion, the results of the conservation carried out are briefly summarized.

In my opinion, the article is not well structured. Two main thematic-methodological areas stand out: one focused on the archival sources and archaeological data verification; the other - on the conservation process of the dwelling architecture. The authors fail to integrate these two blocks harmoniously.

Not being a native speaker myself, I tend not to be judgmental about English language and style deficiencies. Nevertheless, it seems to me that the authors try to translate the Spanish grammatical structure into English. The authors have to check the correctness of using certain words and expressions. An in-depth review of text by a native speaker is highly recommendable.

It would have been preferable to opt for a more straightforward structure: introduction-materials and methods-analysis-results-conclusions.

The block dedicated to conservation could be summarized and presented more synthetically. There are no data on modern conservation principles on archaeological sites and their museumification. Perhaps a figure showing the different sources used and the methodological workflow could help the reader not get lost.

I should emphasize that the materials presented in the article are significant and interested in the scientific community and can be accepted for publication after proofreading.

Author Response

Dear Reviewer 3, 

Thank you for your comments. We have followed all your instructions. We have renamed the titles of some sections to adjust to the required scheme. We have also included new paragraphs explaining in more detail the objectives and methodology. A native English speaker has made a revision of the text. 

Round 2

Reviewer 3 Report

Dear Colleagues, thank you very much for hard work and providing additional evidence in the article. The text became more convincing and clear.